# Bifunctional Non-Canonical Amino Acids: Combining Photo-Crosslinking with Click Chemistry [note 1]

**DOI:** 10.3390/biom10040578

**Published:** 2020-04-10

**Authors:** Jan-Erik Hoffmann

**Affiliations:** Protein Chemistry Facility, Max Planck Institute of Molecular Physiology, Otto-Hahn-Str 11, 44227 Dortmund, Germany; jan-erik.hoffmann@mpi-dortmund.mpg.de

**Keywords:** non-canonical amino acids, genetic code expansion, bifunctionality, photo-crosslinking, click chemistry, protein interaction

## Abstract

Genetic code expansion is a powerful tool for the study of protein interactions, as it allows for the site-specific incorporation of a photoreactive group via non-canonical amino acids. Recently, several groups have published bifunctional amino acids that carry a handle for click chemistry in addition to the photo-crosslinker. This allows for the specific labeling of crosslinked proteins and therefore the pulldown of peptides for further analysis. This review describes the properties and advantages of different bifunctional amino acids, and gives an overview about current and future applications.

## 1. Introduction

The analysis of interactions between proteins and other biomolecules is a highly important task in both structural biology and systems biology. In particular, protein–protein interactions are of great interest for the study of signaling networks and protein complexes [1]. A commonly used tool for this is protein pulldown, where a protein of interest (POI) is captured via an antibody or chemical handle in its native environment, taking binding partners with it. These partners can then be identified via mass spectrometry, immunoblotting, or other methods [2]. Because of recent developments in the field of mass spectrometry, this powerful tool has been successful in identifying many protein–protein interactions [3]. However, the stability of the interaction between the POI and other biomolecules depends on the conditions throughout the pulldown assay, and especially weak and transient interactions are easily lost, leading to false negatives.

Therefore, methods have been developed to stabilize protein interactions, for example via crosslinking with small molecules carrying two chemically reactive groups [4] or photo-activatable bidentate crosslinkers [5]. The formation of a covalent bond between the molecules early in the assay ensures the capture of weaker binding partners. Still, this unspecific crosslinking is usually applied after cell lysis, and some interactions might be lost or there may even be false positives introduced [6]. Ideally, the covalent bond should be formed while the POI is still in its native environment, and it should affect only the POI and its binding partners. In addition, such a tag should be as small as possible so as to reduce the risk of blocking important binding sites [7].

The use of genetic code expansion opens the door to a more specific approach to study protein–protein interactions. Here, a non-canonical amino acid (ncAA) is introduced into a protein at a specific location, carrying a chemical group that is not present in the 20 canonical amino acids [8]. The most common method is amber stop codon suppression, which is therefore the focus of this review, but other ways to expand the genetic code and introduce ncAAs, such as four-base-codons, are also available and can be used instead or even in addition [9]. Amber suppression requires the introduction of a TAG stop codon into the POI gene at the desired site, a tRNA complementary to this stop codon (tRNA_CUA_), the ncAA, and an aminoacyl tRNA synthetase that charges the amber suppressor tRNA with the ncAA. In bacteria and eukaryotic cell cultures, it is common to transiently transfect the cells with a plasmid carrying the genes for synthetase and tRNA_CUA_, and to introduce the POI gene either with another plasmid or through stable transfection or genome editing. Systems where all three components are incorporated in a single plasmid [10] or bacmid [11] have also been described. With all components in place, the ribosome of a host cell can incorporate the ncAA into the POI at the site of the amber codon (Figure 1a) [12].

For the study of protein interactions, the ncAA carries a chemical group that covalently binds to the interaction partner. Photo-crosslinkers that only become reactive upon irradiation with UV-light are especially powerful [13,14]. The main advantage is that the formation of the link can be precisely timed, and is limited to the vicinity of the POI, thereby reducing unspecific binding. Light as a tool provides a high spatial and temporal resolution, and can be compatible with live cells. Introducing such a group by genetic code expansion also allows for precise control over the position within the POI.

The three main classes for ncAA-based photo-crosslinking are benzophenones [15], diazirines [16], and aromatic azides [17]. Benzophenones are commonly irradiated with near-UV light at around 350 nm, and form a diradical that can insert into C–H bonds or other groups, forming a covalent bond with the target molecule [18]. Diazirines are also irradiated at 350 nm, and form a carbene that can also insert into C–H bonds and is even more reactive than the radical formed by benzophenones [19]. Diazirines provide the best time resolution, but can irreversibly react with water to form a ketone. This means that if no biomolecule is nearby, the photo-crosslinker will react with a solvent molecule and stay unlabeled, whereas benzophenones can be reactivated after reacting with water to “enrich” binding partners [20]. On the other hand, diazirines allow for a short flash of light to capture interactions that are present at this exact instance [21]. Aromatic azides are activated at a lower wavelength around 250 nm in order to form a reactive nitrene. While their absorption maximum can be tuned by substitution, in biological experiments, they usually need to be irradiated in the near UV, away from their optimum [22]. At 350 nm, they are therefore less reactive than the other two crosslinkers and usually require longer exposure times.

Photoreactive ncAAs introduced via genetic code expansion have been a powerful tool for over 15 years, and were used to find novel protein–protein interactions and to solve the structure of protein complexes [17,23]. While this method works well for strong interactions and pure samples, one challenge when using universal photo-crosslinker ncAAs is the sample processing prior to the analysis of the interaction partners [24]. Less abundant interactions are likely to be missed when not enriched, and, on the other hand, pulldown strategies via antibodies or affinity tags usually have to be optimized for every POI individually, and somewhat counteract the advantage of a minimal label size [25].

The solution to this problem is the introduction of a second functional group for chemical labeling in the same amino acid [26]. As most photo-crosslinkers are relatively small, a second modification can be tolerated by most aminoacyl-tRNA synthetases, and there are a variety of options for bioorthogonal chemistries. Most prominent are terminal alkynes, which can be labeled with azides via copper-catalyzed click chemistry [27]. This reaction is highly specific even in complex cell lysates, and a variety of azide-reagents are commercially available, such as biotin, fluorophores, and beads [28]. The general workflow is shown in Figure 1b: Within a live cell or complex protein mixture, the POI with the bifunctional ncAA interacts with its native binding partners. Upon UV-irradiation, the photo-crosslinker (blue) reacts with the target and forms a covalent bond that stays intact during processing, such as cell lysis and tryptic digest. The resulting peptide mixture is treated with click chemistry reagents, and only the peptides containing the ncAA are labeled and subsequently pulled out [29]. As all connections are covalent, the washing protocol can be as rigorous as necessary. The purified peptides are then subjected to analysis, such as mass spectrometry, and identified. A similar pipeline can be used for structural studies, where the binding partners are usually known, but the binding sites within the molecule are to be determined.

## 2. Published Bifunctional ncAAs

First introduced in peptide chemistry, bifunctional ncAAs were recently made available for genetic code expansion in live cells. The following paragraph highlights the main representatives of this class and discusses their specific advantages and applications (Figure 2 and Table 1).

### 2.1. AmAzZLys

The first genetically encoded bifunctional ncAA for photo-crosslinking and click chemistry was reported by Yamaguchi et al. from the Sakamoto lab [30]. AmAzZLys (N^ε^-(3-amino-5- azidobenzyloxycarbonyl)-L-lysine; Figure 2) is a derivative of Z-lysine with a benzyl group attached to the lysine side chain via a carbamate. Attached to the benzyl ring in the meta positions are an azide group and an amine. As a pyrrolysine analog, this ncAA can be incorporated with a modified pyrrolysine-synthetase from *Methanosarcina mazei* with the mutations R61K, G131E, Y306A, and Y384F. The aromatic azide acts as a photo-crosslinker, while the amine can be labeled via reductive alkylation. The latter reaction does not meet the strict criteria of click chemistry as it is not fully bioorthogonal, but under the right conditions in a pre-processed sample, the labeling will only occur at the phenylamine and not on other lysines or the N-terminus. The aromatic azide requires a long irradiation time of 30 min in the near UV, which generally limits its use to in vitro experiments. Still, Yamaguchi et al. have demonstrated the successful crosslinking between a nanobody and a receptor, and the subsequent fluorescent labeling of the product. One advantage of AmAzZLys is the flexibility outside of photo-crosslinking chemistry—the azide can be used for an orthogonal one-pot bioconjugation reaction to create a double-labeled nanobody [30,31].

### 2.2. BPKyne

Benzophenone-alanine is a commonly used ncAA for photo-crosslinking, and a bifunctional version carrying a terminal alkyne on the distant end, named BPKyne (4′-ethynyl-p-benzoyl-l-phenylalanine, Figure 2), was first developed for incorporation in peptides by Chen et al. [32]. The Mapp lab has adapted it for genetic code expansion in yeast cells [33], using an engineered *E. coli* tyrosyl synthetase with the mutations Y37G, D182G, and L186A, which was originally developed for benzophenone [34,35]. This incorporation strategy is also compatible with bacterial and mammalian cells. Joiner et al. demonstrated the crosslinking and biotin-pulldown of a transcriptional complex from live cells. The choice of crosslinker and click-handle make this amino acid a strong candidate for experiments in complex biological environments.

### 2.3. DiZASeC

The Chen lab developed a diazirine-based amino acid, DiZASeC (Se-(N-(2-(3-(but-3-yn-1-yl)-3H-diazirine-3-yl)ethyl)propionamide)-3-yl-homoselenocysteine, Figure 2), that, in addition to a terminal alkyne, also features a cleavable selenium-carbon bond—essentially making it a trifunctional ncAA [36]. It is based on an earlier photo-crosslinker by the same lab, which incorporates a selenium atom in the lysine side chain, which undergoes oxidative cleavage upon H_2_O_2_ treatment and separates the crosslinked side chain from the POI [37,38]. The separated chain can be enriched via biotinylation and the isotope labeled at the cleavage site, making it useful for quantitative mass spectrometry. This ncAA has been used to identify novel protease substrates in pathogenic *E. coli* to better understand how it can traverse the host stomach [39]. Despite the modifications to the lysine side chain, DiZASeC is recognized and incorporated by the *Methanosarcina barkeri* pyrrolysyl synthetase with the mutations L274A and C313S, originally developed for another diazirine-lysine [40]. The use of the very reactive diazirine together with the releasable linker and labeling mechanism allows this ncAA to identify less abundant protein interactions that elude other less sensitive methods [41].

### 2.4. PrDiAzK

A similar approach has been used by the Schultz lab who also synthesized a pyrrolysine analog with a diazirine and alkyne group, PrDiAzK (N^ε^-(((3-((prop-2-yn-1-yloxy)methyl)-3H-diazirine-3-yl)methoxy)carbonyl)-L-lysine; Figure 2) [42]. In this case, the bifunctional head is coupled to a native lysine side chain via a carbamate, and the diazirine is placed in closer proximity to the ε-amino group and therefore the protein backbone. The ncAA includes an ester group that makes the side chain less hydrophobic compared with other diazirine-lysines. It is readily incorporated by the *M. mazei* pyrrolysine synthetase Y306A/Y384F mutant that is commonly used for lysine derivatives with large head groups, and is therefore optimized for a variety of host organisms, including mammalian cells [43]. Proof of principle experiments in *E. coli* and human cell culture were conducted and recently it was used to study protein–RNA interactions. This experiment also showed that the position of the diazirine is of great importance for the crosslinking efficiency [44]. In addition, PrDiAzK was incorporated and labeled proteome wide, using sense codon competition. This amino acid is also commercially available (SiChem, Bremen, Germany, Cat# SC-8028).

## 3. Summary and Outlook

The four ncAAs highlighted in this review demonstrate the variety of combinations between photo-crosslinker and click chemistry groups that have been used in different host organisms and for various applications. In particular, DiZASeC from the Chen lab shows great promise and has been shown to outperform conventional photo-crosslinker amino acids. As small variations in the side chain can lead to different results, it is to be expected that further development in this field will lead to even better results. The protein crosslinking community would greatly benefit from the development of more bifunctional amino acids, especially if they become commercially available.

Aside from identifying less abundant and transient binding partners of a single POI, the use of bifunctional ncAAs might also open the door to study whole interactomes. Methods such as stochastic orthogonal recoding of translation (SORT) [45] have been developed to introduce ncAAs into the whole proteome simultaneously. Instead of suppressing a stop codon, the genome is expanded by stochastically incorporating ncAAs in place of certain sense codons and under competition with the native amino acids [46]. Combining this with photo-crosslinkers has the potential to capture interactions throughout the whole proteome, as has been shown by the surrogate-based method, where a photo-crosslinker analog of a canonic amino acid is supplemented and incorporated residue specifically [47,48]. The problem is to isolate and identify the crosslinked proteins, as affinity methods usually only work for individual proteins. Having a click-chemistry group present in every crosslinked peptide, on the other hand, allows for the pulldown of all relevant molecules in a single step, and could potentially generate an interactome fingerprint for each combination of codon and cell states.

## Figures and Tables

**Figure 1 biomolecules-10-00578-f001:**
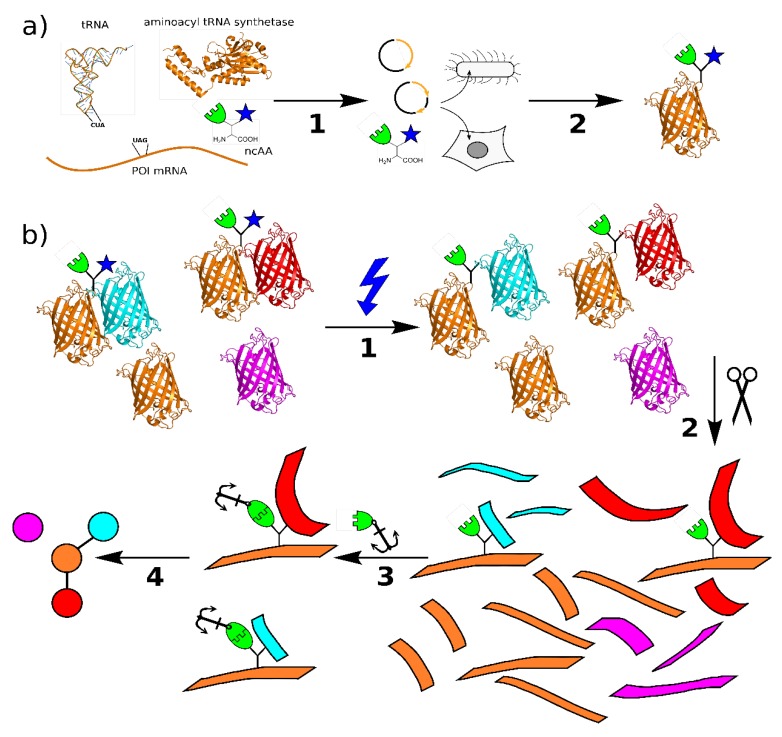
(**a**) Principle of genetic code expansion: The four components required are the mRNA for the target protein with a stop codon at the intended incorporation site, a tRNA complementary to this codon, the non-canonical amino acid, and an aminoacyl-tRNA synthetase that recognizes and connects the tRNA complementary to the TAG stop codon (tRNA_CUA_) and non-canonical amino acid (ncAA). In step 1, these components are introduced into the host organism, e.g., by transient transfection, which then produces the protein of interest (POI) with the ncAA at the selected site in step 2. (**b**) Principle of protein interaction analysis using bifunctional amino acids: The POI incorporating the bifunctional ncAA binds to its native interaction partners in vitro or in vivo. Upon irradiation with UV light (1), a covalent bond is formed between the photo-crosslinker (blue) and the binding partner. The sample is processed by protease digestion (2) and the peptides containing the ncAA are labeled via the click chemistry handle (3). The relevant peptides are enriched by pulldown and analyzed by mass spectrometry. From these data. the protein interactions can be mapped (4) (structures: PDB 1EHZ, 6AAC, and 1GFL).

**Figure 2 biomolecules-10-00578-f002:**
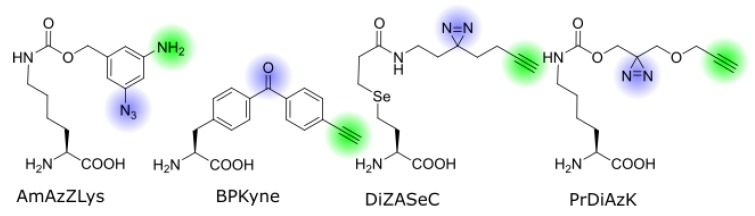
Structures of bifunctional ncAAs discussed in this review. The photo-crosslinker groups and pulldown-handles are highlighted in blue and green, respectively.

**Table 1 biomolecules-10-00578-t001:** Comparison and properties of four different bifunctional ncAAs.

	AmAzZLys	BPKyne	DiZASeC	PrDiAzK
Published	Yamaguci et al., 2016 [30]	Chen et al., 2010 [32]; Joiner et al., 2017 [33]	He and Xie et al., 2017 [36]	Hoffmann et al., 2018 [42]
Pulldown-handle	Phenylamine	Term. alkyne	Term. alkyne	Term. alkyne
Photo-crosslinker	Aromatic azide	Benzophenone	Diazirine	Diazirine
Wavelength used in publication	365 nm	350 nm	350 nm	350 nm
Synthetase	*Mm.* PylRS R61K G131E Y306A Y384F	*Ec.* TyrRS Y37G, D182G, L186A	*Mb.* PylRS L274A, C313S	*Mm*. PylRS Y306A, Y384F

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
