# Peer review of "Bifunctional Non-Canonical Amino Acids: Combining Photo-Crosslinking with Click Chemistry [Author-notes fn1-biomolecules-10-00578]"

_biomolecules, 2020, doi:10.3390/biom10040578_

Round 1

Reviewer 1 Report

This is a nice little review about bi-functional non-canonical amino acids (ncAAs) carrying both a photo-activatable moiety and a chemical anchor for bioorthogonal chemistry. The latter allows enriching the photo-crosslinked peptides before proceeding with mass spec analysis, which reduces the background and greatly facilitates the analysis. The field is evolving very rapidly, and although there are some relatively recent reviews on photo-crosslinking amino acids, there is always need of an updated view on the most recent advances. This manuscript is well organized and clearly written, does not indulge in unnecessary details, makes a nice summary of the crucial aspects of the field, report on the very last literature and is very useful for anyone working with ncAAs and crosslinking. Therefore, I recommend its publication ASAP.

Minor comments

Page 1 line 30. Here the author mentions “chemical crosslinking”. This term is a bit misleading, because there are a bunch on ncAAs designed for chemical crosslinking. He should be more specific. I assume this paragraph is for the chemical crosslinkers that bear two chemical reactive groups joined by a spacer. It would be worth mentioning also the photo-activatable bidentate crosslinkers.

Page 3 line 64. Together with the review von K. Lang also I. Coin, CurrOpChemBiol 2018 “Application of non-canonical crosslinking amino acids to study protein-protein interactions in live cells” should be cited

Page 3 line 64 and following. I recommend revising these few sentences keeping in mind that photo-activatable crosslinker carrying e.g. two benzophenone moieties separated by a spacer exist and they also offer temporal resolution and are compatible with live cells.

Page 3, line 77-79. I am not sure about these statements about azides. When azides are irradiated at short wavelength, crosslinking occurs very rapidly (2-3- minutes) with high yield. The problem is that the wavelength is not biocompatible. Azi is slow when irradiated at long wavelength because in fact the azide has almost no absorbance. These two sentences should be revised.

Page 4, line 117. “aromatic azide requires a long irradiation time of 30 min”. See comment right above.

Page 5, conclusion. Just as a note, maybe it would be good to say that the community wishes that more of these amino acids become commercially available. The bottleneck, why these ncAAs do not find broad application, despite their enormous potential, is that they are not available on the market.

Author Response

Reviewer 1

This is a nice little review about bi-functional non-canonical amino acids (ncAAs) carrying both a photo-activatable moiety and a chemical anchor for bioorthogonal chemistry. The latter allows enriching the photo-crosslinked peptides before proceeding with mass spec analysis, which reduces the background and greatly facilitates the analysis. The field is evolving very rapidly, and although there are some relatively recent reviews on photo-crosslinking amino acids, there is always need of an updated view on the most recent advances. This manuscript is well organized and clearly written, does not indulge in unnecessary details, makes a nice summary of the crucial aspects of the field, report on the very last literature and is very useful for anyone working with ncAAs and crosslinking. Therefore, I recommend its publication ASAP.

I thank the reviewer for this kind and positive summary. I am glad that they agree with my intention to provide a focused overview of a rapidly developing field of chemical biology. The reviewer’s comments are constructive and will improve the manuscript and I hope I have addressed them to their satisfaction.

Minor comments

Page 1 line 30. Here the author mentions “chemical crosslinking”. This term is a bit misleading, because there are a bunch on ncAAs designed for chemical crosslinking. He should be more specific. I assume this paragraph is for the chemical crosslinkers that bear two chemical reactive groups joined by a spacer. It would be worth mentioning also the photo-activatable bidentate crosslinkers.

I clarified the sentence and added Rajagopalan et al., 1994 as a reference for photo-activatable bidentate crosslinking.

Page 3 line 64. Together with the review von K. Lang also I. Coin, CurrOpChemBiol 2018 “Application of non-canonical crosslinking amino acids to study protein-protein interactions in live cells” should be cited

This is a good point; Irene Coin’s review is a great addition to the reference list.

Page 3 line 64 and following. I recommend revising these few sentences keeping in mind that photo-activatable crosslinker carrying e.g. two benzophenone moieties separated by a spacer exist and they also offer temporal resolution and are compatible with live cells.

I added an emphasis that with this method the crosslinking is limited to the vicinity of the POI. I hope that clarifies the difference.

Page 3, line 77-79. I am not sure about these statements about azides. When azides are irradiated at short wavelength, crosslinking occurs very rapidly (2-3- minutes) with high yield. The problem is that the wavelength is not biocompatible. Azi is slow when irradiated at long wavelength because in fact the azide has almost no absorbance. These two sentences should be revised.

This is an important comment and I rewrote the sentence to clarify that the lower reactivity is due to the fact that it is usually irradiated away from the maximum. I also added Geiger et al., 1984 as a reference, who measured several substituted aryl azides to address a comment by reviewer 2. (Even the substituted azides have a maximum far below 350 nm.)

Page 4, line 117. “aromatic azide requires a long irradiation time of 30 min”. See comment right above.

I added a clarification that this refers to the protocol in the near UV.

Page 5, conclusion. Just as a note, maybe it would be good to say that the community wishes that more of these amino acids become commercially available. The bottleneck, why these ncAAs do not find broad application, despite their enormous potential, is that they are not available on the market.

I added the sentence “The protein crosslinking community would greatly benefit from the development of more bifunctional amino acids, especially if they become commercially available.” to the first paragraph of the conclusion.

As a side note, I did try to research further the potential availability of the described amino acids, since reviewer 2 asked for it. The source for PrDiAzK is Sirius Fine Chemical (SiChem, Bremen, Germany) https://shop.sichem.de/en/sc-8028.html). I can verify that, because I ordered a sample and analyzed it by NMR. According to SciFinder, BPKyne might be available Fmoc-protected at Chemieliva (Chongqing, China) (Catalog# CE0099405), but I couldn’t find it anywhere in their shop. Similarly, AmAzZLys seems to be offered by GeorGene Biotech (Shanghai, China) (http://m.georgenebiotech.com/pd.jsp?pid=668), but again I couldn’t find it in their actual shop. This might be due to my limited knowledge of Chinese. Shinsei Chemical Company (Osaka, Japan) offers the Fmoc-protected version of AmAzZLys (https://schem.jp/ComDetail.php?str=02734).

Reviewer 2 Report

After carefully reviewing manuscript biomolecules-749008 entitled “Bifunctional Non-Canonical Amino Acids: Combining Photo-Crosslinking with Click Chemistry” by Jan-Erik Hoffmann, I recommend it for publication after very minor revision.

The study provides a brief but concise review on currently available bifunctional non-canonical amino acids containing photo‐crosslinker and click chemistry groups to explore protein-protein interactions. The study is well structured and easy to read. The figures are well prepared and support and illustrate the facts discussed in the text. Particularly, table 1 is excellent, because it summarizes the review’s content at a glance: photo-crosslinking moieties together with excitation wave lengths are indicated, the click chemistry moieties as well as the incorporation system (which might be augmented by the corresponding mutations, see below) are listed. The facts are supported by the relevant and up-to-date literature. Therefore, the article highly deserves publication in Biomolecules.

However, all facts should be referenced, and in some instances (see my specific comments below), citations are missing. The author should revise the manuscript accordingly before the manuscript can be published.

Specific comments:

lines 40-41: An efficient way to introduce non-canonical amino acids (ncAAs) into proteins is the supplementation method, where an amino acid auxotrophy is supplemented with the corresponding amino acid analog. As a result, the ncAA is incorporated residue-specifically (see e.g. Mishra, et al. (2020) ChemBioChem http://dx.doi.org/10.1002/cbic.201900600; Pickens, et al. (2018) Bioconjug Chem 29(3): 686-701 http://dx.doi.org/10.1021/acs.bioconjchem.7b00633). This approach deserves mentioning and I suggest the author include this method here.

lines 44-46: A system using a single plasmid to encode all components has been described as well: Fladischer, et al. (2019) Biotechnol J 14(3): e1800125 http://dx.doi.org/10.1002/biot.201800125

lines 69-71: The sentence “Benzophenones... target molecule.” needs a reference.

lines 71-72: “...an even more reactive carbene...” more reactive than what?

lines 77-79: The sentences “Aromatic azides... near UV as well.” need a reference/references.

lines 92-102: The claims in this paragraph (after reference [21]) need references.

lines 107, 122, 131, 144: Abbreviations of ncAAS should be accompanied by the trivial/systematic names, e.g. as in the corresponding sections. Obviously, all ncAAs described in the text are shown in Figure 2. I suggest the author refer to these structures upon their first mentioning in the text.

lines 111-121: Claims in the sentences from “As a pyrrolysine analog,...”... double‐labeled nanobody.” need references.

lines 155-156: It would be nice if the commercial source was mentioned. What about the commercial availability of the other ncAAs?

Table 1: Very nice summary of the facts described in the text. I suggest the author includes the mutations of the aminoacyl-tRNA synthetases where known. For instance, “MmPylRS R61K G131E Y306A Y384F” instead of “M.mazei Pyl synthase”, etc. What wavelength is indicated? Please, specify and move closer to the photo-crosslinking group to which is (obviously?) belongs. What does “labeling group” mean? Why does the author not use the same terms as he introduced in Figure 2?

Minor comments and suggestions for amendment:

lines 39, 47: “Amber” should be changed to “amber”

line 43: „Amber‐tRNA“ should be changed to „amber suppressor tRNACUA

line 68: space missing between “benzophenones” and “[11]”

line 76: there is no such English term as “time point”, exchange for “time” or “instance”

Author Response

Reviewer 2

Comments and Suggestions for Authors

After carefully reviewing manuscript biomolecules-749008 entitled “Bifunctional Non-Canonical Amino Acids: Combining Photo-Crosslinking with Click Chemistry” by Jan-Erik Hoffmann, I recommend it for publication after very minor revision.

The study provides a brief but concise review on currently available bifunctional non-canonical amino acids containing photo‐crosslinker and click chemistry groups to explore protein-protein interactions. The study is well structured and easy to read. The figures are well prepared and support and illustrate the facts discussed in the text. Particularly, table 1 is excellent, because it summarizes the review’s content at a glance: photo-crosslinking moieties together with excitation wave lengths are indicated, the click chemistry moieties as well as the incorporation system (which might be augmented by the corresponding mutations, see below) are listed. The facts are supported by the relevant and up-to-date literature. Therefore, the article highly deserves publication in Biomolecules.

However, all facts should be referenced, and in some instances (see my specific comments below), citations are missing. The author should revise the manuscript accordingly before the manuscript can be published.

I thank the reviewer for this positive and very constructive comment. I am happy that the reviewer agrees with the content and structure of the paper. These corrections will improve the quality of the manuscript and I hope I have addressed all of them to their satisfaction.

Specific comments:

lines 40-41: An efficient way to introduce non-canonical amino acids (ncAAs) into proteins is the supplementation method, where an amino acid auxotrophy is supplemented with the corresponding amino acid analog. As a result, the ncAA is incorporated residue-specifically (see e.g. Mishra, et al. (2020) ChemBioChem http://dx.doi.org/10.1002/cbic.201900600; Pickens, et al. (2018) Bioconjug Chem 29(3): 686-701 http://dx.doi.org/10.1021/acs.bioconjchem.7b00633). This approach deserves mentioning and I suggest the author include this method here.

Since all recently published bifunctional ncAAs use the amber suppression method I did focus on this one in the introduction. But I agree with the reviewer that the supplementation/surrogate method is well established in the field of protein crosslinking (particularly using diazirines) and should be mentioned. However, I think a better place to do so would be in the outlook, where I discuss the proteome-wide residue specific incorporation of ncAAs. I therefore added a sentence to compare the established supplementation methods with the potential of using bifunctional ncAAs with SORT and linked the references suggested by the reviewer. I hope is alright, but if the reviewer prefers to have this point discussed already in the introduction I could add a sentence there.

lines 44-46: A system using a single plasmid to encode all components has been described as well: Fladischer, et al. (2019) Biotechnol J 14(3): e1800125 http://dx.doi.org/10.1002/biot.201800125

This is a good point and I added this fact along with the reference. I also added a reference to Koehler et al., where tRNA, synthetase and POIs were incorporated in a single bacmid for insect cell expression.

lines 69-71: The sentence “Benzophenones... target molecule.” needs a reference.

I added Dorman et al., 1994 as a reference, who explore the crosslinking mechanism.

lines 71-72: “...an even more reactive carbene...” more reactive than what?

I clarified that the comparison is with the radical from the benzophenone. I provided Sakurai at al., 2014 as a reference, who performed a comparative study on the three common photo-crosslinkers.

lines 77-79: The sentences “Aromatic azides... near UV as well.” need a reference/references.

I added Geiger et al., 1984 as a reference who tested several substituted aryl azides. To address a comment by reviewer 1, I also rewrote the sentence to clarify that even the substituted aryl azides need to be activated away from their maximum in biological samples, explaining the lower reactivity.

lines 92-102: The claims in this paragraph (after reference [21]) need references.

I linked Fu et al., 2014 who give a protocol for click pull-down from cell lysate. I also added a reference to Haberkant et al., 2014. It is a review about bifunctional lipids, a field where this approach is well established and it describes the copper-click pull-down method from biological environments after crosslinking in detail. I did not cite the actual bifunctional ncAAs in this paragraph, since they will be described in detail in the following section; including references to their specific pull-down protocols.

lines 107, 122, 131, 144: Abbreviations of ncAAS should be accompanied by the trivial/systematic names, e.g. as in the corresponding sections. Obviously, all ncAAs described in the text are shown in Figure 2. I suggest the author refer to these structures upon their first mentioning in the text.

I added the systematic name of each ncAA in brackets after their first mention, accompanied by a reference to figure 2.

lines 111-121: Claims in the sentences from “As a pyrrolysine analog,...”... double‐labeled nanobody.” need references.

Most of this paragraph refers to the original paper describing AmAzZLys (Yamaguchi et al., 2014), so I referenced it again at the end of the paragraph for clarification. I also added another reference Schumacher et al., 2018 that discusses the possibility of using it for nanobody labeling.

lines 155-156: It would be nice if the commercial source was mentioned. What about the commercial availability of the other ncAAs?

The source for PrDiAzK is Sirius Fine Chemical (SiChem, Bremen, Germany) https://shop.sichem.de/en/sc-8028.html). I can verify that, because I ordered a sample and analyzed it by NMR. According to SciFinder, BPKyne might be available Fmoc-protected at Chemieliva (Chongqing, China) (Catalog# CE0099405), but I couldn’t find it anywhere in their shop. Similarly, AmAzZLys seems to be offered by GeorGene Biotech (Shanghai, China) (http://m.georgenebiotech.com/pd.jsp?pid=668), but again I couldn’t find it in their actual shop. This might be due to my limited knowledge of Chinese. Shinsei Chemical Company (Osaka, Japan) offers the Fmoc-protected version of AmAzZLys (https://schem.jp/ComDetail.php?str=02734).

I would like to ask the editors for their opinion on including commercial sources in this review. Like reviewer 1 notes, there is definitely an interest in the community to learn about the availability of these tools. On the other hand, I would not like to provide undue advertisement for a single company. As a disclaimer, while I am co-author on the publication describing the original synthesis of PrDiAzK, I do not hold a patent on it or have any other financial interest in the company that sells it. For now, I did add the source in brackets in the review, but I would be fine if the editors decide to remove the sentence altogether. I did not add the Chinese companies, since I cannot verify if the actually have the amino acids available. I could add Shinsei, but since they sell only the protected version it is not immediately suited for the purpose described in the review.

Table 1: Very nice summary of the facts described in the text. I suggest the author includes the mutations of the aminoacyl-tRNA synthetases where known. For instance, “MmPylRS R61K G131E Y306A Y384F” instead of “M.mazei Pyl synthase”, etc. What wavelength is indicated? Please, specify and move closer to the photo-crosslinking group to which is (obviously?) belongs. What does “labeling group” mean? Why does the author not use the same terms as he introduced in Figure 2?

I rearranged the table according to the reviewer’s suggestions, clarified the titles and added the synthetase mutations. In case of the BPKyne the mutations weren’t clearly stated in the publication, but I assume they are derived from Chin et al., 2003 (this paper is cited in their previous publication about BpA that seems to use the same synthetase). I added this reference and the mutations to the paragraph about BPKyne.

Minor comments and suggestions for amendment:

lines 39, 47: “Amber” should be changed to “amber”

This is an interesting point. Since “amber” is derived from a surname I originally capitalized it. But I acknowledge that the minor spelling is dominant in the literature and changed it accordingly.

line 43: „Amber‐tRNA“ should be changed to „amber suppressor tRNACUA

line 68: space missing between “benzophenones” and “[11]”

line 76: there is no such English term as “time point”, exchange for “time” or “instance”

I have incorporated these corrections.